# Antibacterial and Antispore Activities of Isolated Compounds from *Piper cubeba* L.

**DOI:** 10.3390/molecules24173095

**Published:** 2019-08-26

**Authors:** Fatimah Alqadeeri, Yaya Rukayadi, Faridah Abbas, Khozirah Shaari

**Affiliations:** 1Laboratory of Natural Products, Institute of Bioscience, Universiti Putra Malaysia, Serdang 43400, Selangor, Malaysia; 2Department of Food Science, Faculty of Food Science and Technology, Universiti Putra Malaysia, Serdang 43400, Selangor, Malaysia; 3Department of Chemistry, Faculty of Science, Universiti Putra Malaysia, Serdang 43400, Selangor, Malaysia

**Keywords:** antibacterial activity, antispore activity, *Bacillus* sp, *P. cubeba* L., β-asarone, asaronaldehyde

## Abstract

*Piper cubeba* L. is the berry of a shrub that is indigenous to Java, Southern Borneo, Sumatra, and other islands in the Indian Ocean. The plant is usually used in folk traditional medicine and is an important ingredient in cooking. The purpose of this study was to isolate and purify the bioactive compounds from *P. cubeba* L. fractions. In addition, the isolated compounds were tested for their antibacterial and antispore activities against vegetative cells and spores of *Bacillus cereus* ATCC33019, *B. subtilis* ATCC6633, *B. pumilus* ATCC14884, and *B. megaterium* ATCC14581. The phytochemical investigation of the DCM fraction yielded two known compounds: β-asarone (1), and asaronaldehyde (2) were successfully isolated and identified from the methanol extract and its fractions of *P. cubeba* L. Results showed that exposing the vegetative cells of *Bacillus* sp. to isolated compounds resulted in an inhibition zone with a large diameter ranging between 7.21 to 9.61 mm. The range of the minimum inhibitory concentration (MIC) was between 63.0 to 125.0 µg/mL and had minimum bactericidal concentration (MBC) at 250.0 to 500.0 µg/mL against *Bacillus* sp. Isolated compounds at a concentration of 0.05% inactivated more than 3-Log_10_ (90.99%) of the spores of *Bacillus* sp. after an incubation period of four hours, and all the spores were killed at a concentration of 0.1%. The structures were recognizably elucidated based on 1D and 2D-NMR analyses (1H, 13C, COSY, HSQC, and HMBC) and mass spectrometry data. Compounds 1, and 2 were isolated for the first time from this plant. In conclusion, the two compounds show a promising potential of antibacterial and sporicidal activities against *Bacillus* sp. and thus can be developed as an anti-*Bacillus* agent.

## 1. Introduction

*Piper cubeba* L. is the berry of a shrub that is indigenous to Java, Southern Borneo, Sumatra, and other islands in the Indian Ocean. Various types of these plants grow in wild forests while others are cultivated in coffee plantations in Java. *Piper* is a genus of the Piperaceae family and is known to have no less than 700 species across the world. In tropical regions, members of the genus *Piper* are utilized in a variety of ways, including as foods and spices, oils, and fish poison [1].

Recently, a technique has been developed for isolating and purifying bioactive compounds extracted from plants [2]. This modern technique is able to precisely isolate, separate, and purify compounds and has advantages that are similar to those of many advanced bioassays. The utilization of appropriate methods is crucial when examining bioactive compounds in order to be able to quickly and accurately screen source material for bioactivities such as antioxidant, antibacterial, or cytotoxicity [2]. The selection and collection of plant materials are important steps in isolating and characterizing bioactive phytochemicals. This step is followed by gathering ethnobotanical information to distinguish possible bioactive molecules. Several solvents can be used to obtain plant extracts, and active compounds are then isolated and purified in order to determine the active compounds that are responsible for the bioactivity. Column chromatographic techniques (CC) can be used to isolate and purify bioactive compounds. Advanced and innovative instruments such as high-pressure liquid chromatography (HPLC) allow for the faster purification of bioactive molecules.

Several spectroscopic techniques such as UV-Visible, infrared (IR), nuclear magnetic resonance (NMR), and mass spectroscopy can be utilized to determine the purified compounds [2]. Paper thin-layer and column chromatographic methods have been widely used to isolate and purify many bioactive molecules. Thin-layer chromatography (TLC) and column chromatography are frequently employed due to their effectiveness, low cost, and accessibility in various stationary phases. Silica, alumina, cellulose, and polyamide are very useful for separating phytochemicals. Due to the large quantities of the complex phytochemicals present in plant materials, it is rather difficult to develop an excellent separation technique. Thus, it is necessary to utilize multiple mobile phases to increase polarity in order to obtain a high-quality separation.

Thin-layer chromatography has always been used to analyze compounds from fractions via column chromatography. Several available analytical tools have been used with silica gel column chromatography and thin-layer chromatography (TLC) to separate bioactive molecules. The objective of this study was to isolate and purify the bioactive compounds from *P. cubeba* L. fractions that have antibacterial and antispore activities against the vegetative cells and spores of *Bacillus cereus* ATCC33019, *B. subtilis* ATCC6633, *B. pumilus* ATCC14884, and *B. megaterium* ATCC14581.

## 2. Results and Discussion

### 2.1. Structure Elucidation

Two compounds were isolated from the bioactive dichloromethane fraction. The structures of the compounds (Figure 1) were elucidated using ^1^H-, ^13^C-NMR, and mass spectral data as well as through a comparison with the values published in the literature. The identified compounds were β-asarone (1) [3,4] and asaronaldehyde (2) [4] (see Appendix A) This is the first time compounds 1 and 2 had been isolated from *P. cubeba* berries. The mass spectra for compounds 1, and 2 were measured with a direct injection probe mass spectrometer (DIP-MS) on a gas chromatography-mass spectrometry(GC-MS) Shimadzu Qp-2010 (Kyoto, Japan).

Compound 1 has been previously identified in various plants, for instance, in the leaves and inflorescences of *Helichrysum arenarium* L. Moench [5], rhizomes of *Acorus calamus* and *Acorus gramineus* [3], *Acorus tatarinowii* Schot rhizomes, and *Guatteria gaumeri* Greenman [6]. Compound 1 was identified on the TLC plate as a purple-blue spot after detection with a 365 nm UV lamp at an R_f_ value of 0.64, which is in accordance with the report made by Syarif et al. [7]. Compound 1 (10.4 mg) was obtained as a yellow tablet crystal with a melting point of 99–89 °C. The DIP-MS data showed a molecular ion peak at *m/z* 208 [M]^+^ (100), 193 (47), 177 (3), 165 (27), 162 (12), 150 (6), 137 (18), 105 (4), 91 (11), 77 (7), 69 (15), and 53 (2), which corresponded with the molecular formula C_12_H_16_O_3_ (Figure 2). The ^1^H-NMR (CDCl_3_, 500 MHz) spectra analysis (see Appendix A) had δ values of δ 6.49 (1H, s, H-3), 6.94 (1H, s, H-6), 6.66 (1H, s, H-7), and 6.13 (1H, m, H-8), which indicate the presence of olefinic protons, a methyl group attached with olefinic bond 1.83 (3H, dd, H-3′), three methoxy protons singlet at δ 3.86 (3H, s, H-10), 3.88 (3H, s, H-11), and 3.82 (3H, s, H-12); the ^13^C-NMR (CDCl_3_, 125 MHz) spectrum showed twelve carbon signals (see Appendix A), which corresponded to the suggested molecular formula of the compound, C_12_H_16_O_3_, at δ ppm: δ, 143.3 (C-1), 148.7 (C-2), aromatic carbons δ 97.8 (C-3), 118.9 (C-5), 150.6 (C-4), 114.6 (C-6), olefinic carbons 125.0 (C-7), 124.4 (C-8), 18.8 (C-9), methoxy carbons 56.4 (C-10), 56.1 (C-11), and 56.7 (C-12) (3× -OCH_3_).

The connectivities observed between proton H-3 resonated at δ 6.49 and carbons C-2 (δ 148.7), C-4 (δ 150.6), C-1 (δ 143.3), and C-5 (δ 118.9), and those between proton H-6 resonated at (δ 6.94) and carbons C-1 (δ 143.3), C-5 (δ 118.9), C-7 (δ 125.0), C-2 (δ 148.7), and C-4 (δ 150.6) further confirmed these chemical shift assignments through their HMBC experiment (see Appendix A). The assignment of the OCH_3_ group to carbons C-1/2/4 was established by the connectivity observed in the HMBC spectrum (Appendix A) between the OCH_3_ proton and the carbons resonated at δ C-1 (δ 143.3), C-2 (δ 148.7), and C-4 (δ 150.6). Further proton and carbon correlations can be seen in Appendix A under the HMBC ^2^*J* and HMBC ^3^*J* headings. Based on the spectral data, the compound was identified as β-asarone (2,4,5-trimethoxybenzene; Figure 1). Its identity was confirmed by comparing the data with those published in the literature [3,4]. This is the first report of this compound being isolated from *P. cubeba* berries.

Compound 2 has been previously identified in several plants, for example, in the rhizomes of *Boesenbergia thorelii* [8], rhizomes of *Acorus calamus* [9], and *Acorus tatarinowii* Schott rhizomes [4]. Compound 2 (10.8 mg) was obtained as a colorless tablet crystal with a melting point at 110–112 °C (Lit. 110–112 °C) [4]. The DIP-MS data showed a molecular ion peak at *m*/*z* 196 [M]^+^ (100), 181 (49), 150 (32), 125 (25), 110 (15), 95(7), 79(5), and 69(10), which corresponded with the molecular formula C_10_H_12_O_4_ (Figure 3). The ^1^H-NMR (CDCl_3_, 500 MHz) spectra analysis of compound 2 (see Appendix A) showed a singlet at δ 10.32 s, 1H (–CH=O) H-7, which suggests that it contains an aldehydic proton in its molecule. The presence of three singlets at 3.97 (3H, s, H-8), 3.92 (3H, s, H-9), and 3.88 (3H, s, H-10), indicated the presence of three methoxy groups. In addition, the presence of two singlets at 6.49 (1H, s, H-3) and 7.33 (1H, s, H-6) ppm suggests the presence of a tetra substituted aromatic ring in the molecule. The ^1^H-NMR spectrum of compound 2 is identical to the published spectra of 2,4,5-trimethoxybenzaldehyde [4,10].

The ^13^C-NMR (CDCl_3_, 125 MHz) spectrum showed ten carbon signals (see Appendix A), which corresponded to the suggested molecular formula of the compound, C_10_H_12_O_4_ at δ ppm: δ 96.1 (C-3), 109.2 (C-6), 188.3 (C-7), methoxy carbons 56.46 (C-8), 56.49 (C-9), 56.56 (C-10; 3x-OCH3), 117.5 (C-1), 143.5 (C-2), 158.9 (C-4), and 156.8 (C-5). The connectivities observed (see Appendix A) between proton H-3 resonated at δ 6.49 and carbons C-2 (δ143.5), C-4 (δ158.9), C-1 (δ117.5), and C-5 (δ156.8) while those between proton H-6 resonated at (δ7.33) and carbons C-1 (δ117.5), C-5 (δ156.8), and C-7 (δ188.3) further confirmed these chemical shift assignments through their HMBC experiment (see Appendix A). The assignment of the OCH3 group to carbons C-2/4/5 was established by the connectivity observed in the HMBC spectrum (see Appendix A) between the OCH3 proton and the carbons resonated at δ C-2 (δ143.5), C-4 (δ158.9), and C-5 (δ156.8). Further proton and carbon correlations can be seen in Appendix A under the HMBC ^2^J and HMBC ^3^J headings. Based on this spectral data, the compound was identified as asaronaldehyde (2,4,5-trimethoxybenzaldehyde; Figure 1). Its identity was confirmed by comparing the data with those published in the literature [4]. This is the first report of this compound being isolated from *P. cubeba* berries.

### 2.2. Antibacterial Activity of Bioactive Compounds

#### 2.2.1. Disc Diffusion Assay (DDA)

There are a few reports concerning the susceptibility of *B. cereus*, *B. subtilis*, *B. pumilus,* and *B. megaterium* to a single antimicrobial compound isolated from this plant. The disc diffusion assay of bioactive compounds against *Bacillus* sp. was evaluated. The data for disc diffusion assay (DDA) are summarized in Table 1. The table shows that the compounds isolated from *P. cubeba* L. at a concentration of 0.1% had significant antimicrobial activity against the tested strain. The compounds isolated from *P. cubeba* L demonstrated a broad-spectrum activity against all tested bacteria. The inhibition zone ranged between 7.2 ± 0.2 to 9.3 ± 0.5 mm. The results of the disc diffusion assay showed the largest inhibition zone for β-asarone in contrast to asaronaldahyde.

The inhibition zone produced by chlorhexidine CHX (10 mg/mL) as the positive control was 11.00 mm, and DMSO 10%, which was the negative control, did not inhibit the growth of the tested strain. The exposure of *B. cereus*, *B. subtilis*, *B. pumilus*, and *B. megaterium* to β-asarone resulted in inhibition zones of 9.31 ± 0.50, 8.51 ± 0.50, 8.51 ± 0.50, and 8.15 ± 0.50 mm, respectively. However, the inhibition zones produced by asaronaldehyde were 8.00 ± 0.00, 8.81 ± 0.24, 7.81 ± 0.22, and 7.21 ± 0.21 mm. A larger inhibition zone indicates the higher antibacterial activity of the compounds exposed to the tested *Bacillus* species. Generally, β-asarone is known to be the most abundant component of *A. calamus*, and among others has been shown to have biological functions as an antibacterial [3].

The results in this study are supported by those reported by McGaw et al. [3]; purified β-asarone isolated from the rhizome *A. calamus* inhibited the growth of bacteria Gram-positive *Bacillus subtilis* ATCC6051, *Staphylococcus aureu*s ATCC12600, Gram-negative *Escherichia coli* ATCC11775, and *Klebsiella pneumonia* ATCC13883. However, the concentrations used in these researchers were very high in comparison to the concentration used in the present study. The antimicrobial strength of β-asarone can be attributed to its mechanism of action, which could be similar to the membrane disruption of susceptible test organism [11]. Generally, bacteria species exhibit a larger inhibition zone between 11.1 ± 0.11 mm and 11.6 ± 0.30 mm. Asaronaldehyde and β-asarone showed a lower inhibition zone in the range of 7.21 ± 0.21 to 9.31 ± 0.50 mm when compared to that of the control positive. Tippayatum and Chonhenchob [12] reported that thymol, eugenol, and nisin showed the smallest inhibition zones of 8.0 ± 0.8, 7.0 ± 0.4, and 7.2 ± 0.4 mm for *B. cereus*, which were similar to the results obtained in the present study. The antibacterial activity of the isolated compounds was lower than that of the crude extract and its fractions. This may again be partly attributed to the lability of the isolated compound. As they make up such a small part of the overall composition, their identification was not attempted. Several compounds appeared to be the major component of the crude extract. However, if it were the major antibacterial compound, it would be expected to have a much lower minimum inhibitory concentration (MIC) value. This was not the case, so other minor compounds may well have significantly higher antibacterial activity, contributing to a greater proportion of the overall activity even though they are not present in such high quantities.

#### 2.2.2. Minimum Inhibitory Concentration (MIC) and Minimum Bactericidal Concentration (MBC) of the Isolated Compounds

The susceptibility of bioactive compounds against *B. cereus*, *B. subtilis*, *B. pumilus,* and *B. megaterium* was determined in terms of minimum inhibitory concentration (MICs), and minimum bactericidal concentration (MBCs). All strains showed the sensitivity of β-asarone and asaronaldehyde. The compounds had lower MIC and MBC than the *P. cubeba* L. extract and its fractions. This finding provides strong evidence that β-asarone and asaronaldehyde are responsible for the antibacterial activity of the *P. cubeba* L. extract. The MIC values of the compounds are presented in Table 2. Asaronaldehyde and β-asarone have the same MIC of 125 µg/mL against *B. cereus*, *B. subtilis*, *B. pumilus*, and *B. megaterium*.

All tested *Bacillus* sp. showed a higher MBC ranging between 250 to 500 µg/mL when compared to those of the MIC. There have been several reports concerning the susceptibility of *B. cereus* to a single antimicrobial compound isolated from plants. Rukayadi et al. [13] reported that macelignan significantly inhibited the growth of vegetative cells of *B. cereus* with a much smaller MIC of 4 µg/mL, and also had strong bacterial static (MBC) activity against *B. cereus* vegetative cells with a MBC of 8 µg/mL. Additionally, Mokbel and Hashinaga [14] reported that β-sitosterol and oleic acid at the concentration of 300 and 250 µg/mL inhibited the growth of vegetative cells of *B. cereus*; these concentrations were much higher than the values obtained in the present study.

Similar results were reported for the crude extracts of different plants containing β-sitosterol against various microorganisms [15,16,17]. However, the concentrations used in these studies were much higher than the concentration used in the present results. The leaf and rhizome of *Acorus calamus* have been shown to have antibacterial activity. *A. calamus* rhizomes exhibited strong antibacterial activity against *Pseudomonas aeruginosa*, *S. aureus*, and *B. subtilis* with a MIC of 0.25 [18]. Mycobacterium sp. and *B. subtilis* are both susceptible to calamus oil [19]. Dilika et al. [20] reported that the linoleic acid isolated from the leaves of *Helichrysum pedunculatum* has the ability to inhibit the growth of all Gram-positive bacteria such as *B. cereus* and *B. subtilis* with a MIC of 10 and 10 µg/mL, respectively, These MIC were higher than the current study.

### 2.3. Antispore Activity

Alternative methods for controlling bacterial endospore contamination are urgently needed in various industries and applications. In recent years, more attention has been given to natural products such as essential oils for their sporicidal activity. There have been a few reports concerning the susceptibility of *Bacillus* sp. to a single antispore compound isolated from plants. In the current study, two compounds, i.e., β-asarone and asaronaldehyde were investigated for their abilities to reduce the viability of *B. subtilis*, *B. cereus*, *B. pumilus*, and *B. megaterium* spores.

#### 2.3.1. Sporicidal Activities of β-Asarone against Spores of *Bacillus subtilis*, *Bacillus cereus*, *Bacillus pumilus*, and *Bacillus megaterium*

The sporicidal activities of varying concentrations of β-asarone (62.5, 125, 250, 500, and 1000 µg/mL) against *B. cereus*, *B. subtilis*, *B. pumilus*, and *B. megaterium* are shown in Table 3, Table 4, Table 5 and Table 6). A marked reduction in the spore densities of all *Bacillus* was achieved when the spores were exposed to 1000 µg/mL β-asarone, which was a reduction of ˃3 Log_10_ units in the number of spores/mL. This compound was able to reduce the number of viable *B. cereus*, *B. subtilis*, *B. pumilus*, and *B. megaterium* spores to 2.74 ± 0.04, 2.81 ± 0.05, 3.43 ± 0.05, and 3.43 ± 0.05, respectively, after being exposed to a concentration of 500 µg/mL for one h. The death of all spores of *B. cereus*, *B. subtilis*, *B. pumilus*, and *B. megaterium* was achieved when the concentration of β-asarone was increased to 1000 µ/mL. A higher concentration of the compound resulted in a stronger effect.

No significant difference in the results was observed when 125 and 250 µg/mL of β-asarone were used in comparison to other concentrations. The results of the present study support the findings of earlier work which showed that essential oils such as bergamot, cardamom, clove bud, eucalyptus blue gum, juniper leaf, laurel leaf, lemongrass, palmarosa, peppermint, pine, tea tree, thyme, and yarrow were able to reduce the number of viable *B. subtilis* spores with a mean Log_10_ reduction of spores of between 0.7 ± 0.18 and 3.12 ± 0.21 [21].

Tripoli et al. [22] reported that purified oleuropen isolated from olive extract inhibited the germination and outgrowth of *B. cereus* spores, which was consistent with the findings of this study. Hamad et al. [23] tested the essential oils from the leaves of *Syzygium polyanthum* and *Syzygium aromaticum* against four food-borne microorganisms including *B. subtilis*, *E. coli*, *S. typhimurium* and *S. aureus*. The major ingredient in the essential oil of *S. polyanthum* is cis-4-decanal (43.489%), and that of *S. aromaticum* is p-eugenol (75.190%). The use of essential oils from *S. polyanthum* resulted in the greatest inhibition of the growth of *B. subtilis* with a MIC value of 31.25 µg/mL, while the values for the other isolated bacteria were ˃1000 µg/mL; the essential oil from *S. aromaticum* had a greater and visible inhibitory activity against *B. subtilis* with a MIC value of 31.25 µg/mL, while the value for the other bacteria examined in this study was 250 µg/mL. However, both of the essential oils were not able to inhibit the growth of *E. coli*. The activities of the essential oils from *S. polyanthum* and *S. aromaticum* are by virtue of their major chemical constituents, i.e., cis-4-decanal, aldehydes, and eugenol. These results are consistent with the findings of the present study.

Devi et al. [9] assessed the antimicrobial activities of the purified and refined fraction of β-asarone obtained from the column chromatographic preparation of a crude methanol extract of *Acorus calamus* rhizomes on various microorganisms, including bacteria, yeasts, and filamentous fungi. The fraction demonstrated high activity against filamentous fungi such as *Trichophyton rubrum*, *Microsporum gypseum*, and *Penicillium marneffei* with MIC values of 0.2, 0.2, and 0.4 mg/mL, respectively. Its activities against yeasts, i.e., *Candida albicans*, *Cryptococcus neoformans,* and *Saccharomyces cerevisiae* were moderate with a MIC ranging between 0.1–1 mg/mL, while the activity against bacteria was low with a MIC of between 5–10 mg/mL.

Burt [24] reported that thymol and carvacrol are phenolic compounds present in the essential oil fraction from the *Origanum* and *Thymus* genera. These compounds have the ability to prevent the growth of fungi and bacteria, including foodborne pathogens. Due to their hydrophobicity, these compounds probably dissolve in the hydrophobic domain of the cytoplasmic membrane between the lipid acyl chains and cause alterations in membrane permeability and the activity of enzyme systems [24]. Singh et al. [25] examined the activity of active compounds such as alkaloids, flavonoids, saponins, tannins, and triterpenoids present in *P. cubeba* L. berries and found that they were able to prevent the growth of *B. cereus* and *B. subtilis* with inhibition zones ranging between 20.6 to 30.5 mm.

According to Friedman et al. [26], oregano oil has strong antibacterial activity against the tested organisms. The inhibitory effect of the oil is especially notable against the vegetative cells of *B. cereus*; at higher concentrations, the oil is able to completely inhibit the growth of the *B. cereus* spores. A comparison of the Log_10_ colony forming unity (CFU) values and the calculated percentage of inhibition at different concentrations showed that the activity against vegetative cells of *B. cereus* was stronger than that against the spores of *B. cereus*. Friedman et al. [26] reported that cinnamon oil, oregano oil, thyme oil, carvacrol, (*S*)-perillaldehyde, 3,4-dihydroxybenzoic acid (*b*-resorcylic acid), and 3,4-dihydroxyphenethylamine had exceptional activities against the vegetative cells and spores of *E. coli* NCTC1186, *S. aureus* ATCC12715, and *B. cereus.* Previous studies have reported the same findings where macelignan significantly inhibited growth with a MIC of 4µg/mL on the vegetative cells of *B. cereus* and completely killed them at a MBC of 8 µg/mL. Macelignan has the ability to deactivate more than 3-Log_10_ of spore/mL [27]. These results support the findings of the present results.

#### 2.3.2. Sporicidal Activities of Asaronaldehyde Against Spores of *Bacillus subtilis*, *Bacillus cereus*, *Bacillus pumilus*, and *Bacillus megaterium*

The sporicidal activity of varying concentrations (62.5, 125, 250, 500, and 1000 µg/mL) of asaronaldehyde against *B. cereus*, *B. subtilis*, *B. pumilus*, and *B. megaterium* is shown in Table 7, Table 8, Table 9 and Table 10. A marked reduction in the number of spores/mL of >3 Log_10_ units for all *Bacillus* was achieved at an asaronaldehyde concentration of 500 µg/mL. Asaronaldehyde was able to reduce the number of viable *B. cereus* and *B. megaterium* spores to 3.40 ± 0.21 and 3.90 ± 0.08, respectively, after a 1-h exposure to a concentration of 500 µg/mL. The number of viable *B. subtilis* and *B. pumilus* spores was reduced to 3.58 ± 0.07 and 3.60 ± 0.6 after a 4-h exposure to a concentration of 500 µg/mL A higher concentration of the compound resulted in the lower logarithm of the remaining spores. All spores of *B. cereus*, *B. subtilis*, *B. pumilus*, and *B. megaterium* were killed when treated with 1000 µg/mL asaronaldehyde for 1, 2, 3, and 4 h, respectively. There was no significant difference in the results when using 62.5, 125, and 250 µg/mL of asaronaldehyde. The results of this study were consistent with those of earlier research that showed that the ethanol extract of *Torilis japonica* had antimicrobial properties against the spores of *B. subtilis* and was able to reduce the number of viable *B. subtilis* spores.

Toroglu [28] reported that the essential oils extracted from Sardinian thymus species and compounds such as a-terpineol, a-pinene, *p*-cymene, *g*-terpinene, linalool, carvacrol, and thymol had antimicrobial activities against *E. coli* ATCC25922, *E. coli* O157:H7 ATCC35150, *P. aeruginosa* ATCC27853, *S. aureus* ATCC25923, *S. epidermidis* ATCC12228, *E. faecalis* ATCC29212, *Yersinia enterocolitica* ATCC9610, *Candida albicans* ATCC10231 (Difco laboratories); *B. cereus* ATCC11778, *Listeria monocytogenes* ATCC7644, *Salmonella typhimurium* ATCC14028, and *Sacc. cerevisiae* ATCC9763 (Oxoid). The results of their study showed that the oil of T. herbal-barona ‘b’ had the strongest antimicrobial activity against all tested strains, as demonstrated by the higher percentage of isolates with MBCs equal to or lower than 450 mg/mL, and the compounds showing MBCs higher than 900 mg/mL against all strains.

## 3. Materials and Methods

### 3.1. Materials

Analytical grade solvents, i.e., *n*-hexane, chloroform, ethyl acetate, methanol, butanol, dichloromethane, and acetone were purchased from R&M Chemical (Selangor, Malaysia). The solvents were used in the extraction and chromatography processes. Deionized water was purified using a Milli-Q purification system (supplied by Milipore, Bedford, MA, USA). The deuterated solvents used in the NMR analysis were chloroform-d1 (CDCl_3_), supplied by Merck (Darmstadt, Germany). Sulfuric acid (H_2_SO_4_) was purchased from Sigma Chemical Co. (St. Louis, MO, USA). Media used for the antibacterial activities included: Mueller–Hinton broth (MHB), Mueller–Hinton agar (MHA), and nutrient agar (NA), which were provided by Oxoid Ltd. (Basingstoke, UK). Chlorhexidine (CHX) was purchased from Sigma Aldrich Co. (St. Louis, MO, USA). *Bacillus cereus* ATCC33019, *Bacillus subtilis* ATCC6633, *Bacillus pumilus* ATCC14884, and *Bacillus megaterium* ATCC14581 were obtained from American Type Culture Collection (Gaithersburg, MD, USA).

### 3.2. Plant Materials

The dried *P. cubeba* L. (Piperaceae) berries employed in this study were obtained from a market selling traditional herbs in Pasar Baru Bandung, Indonesia. The *P. cubeba* L. was gathered in April 2015 in a plantation in Jatiroto, Temanggung, and Central Java, Indonesia. The Department of Biology, Institut Teknologi Bandung (Indonesia) authenticated the berries on the basis of the Flora of Java (Backer and Van de Brink, 1968). A voucher specimen (HBG10PC01) was stored at the Herbarium Bandungense. The gathered material was air-dried and put in storage at the Laboratory of Natural Products, Institute of Bioscience (IBS), University of Putra Malaysia (UPM). A powerful heavy duty blender (Waring, Model 32 BL80, New Hartford, NY, USA) was used to pulverize the dried berries into a fine powder. The powdered *P. cubeba* L. sample was stored in an airtight polyethylene plastic bag and put in storage in a −80 °C fridge.

### 3.3. Extraction Procedure

The extraction of *Piper cubeba* L. was done utilizing the soaked method illustrated by Rukayadi et al. [29]. The organic solvent used in the extraction of *P. cubeba* L. was absolute methanol (R & M Chemicals, 99.8%). One hundred grams of dried *P. cubeba* L. berries were ground to obtain a bristly powder. Sample extraction was performed using 400 mL of solvent at room temperature and 48 h of conventional shaking. Filtration of the plant extract was done using Whatman filter paper size No. 2 (Whatman International Ltd., Middlesex, UK). Following this, the extracts were concentrated using a rotary vacuum evaporator (Heidolph VV2011, Schwabach, Germany) at 40 °C for 3–4 h to obtain a methanol extract of the dried *P. cubeba* L. berries. The temperature of the rotary evaporator was increased to 85 °C for 2 × 30 s at the end of the extraction process to ensure that the extract was methanol-free [30]. Finally, the extracts were freeze-dried for 48 h to eliminate water.

### 3.4. Liquid–Liquid Partition of P. cubeba L. Extract

A liquid–liquid partition of the crude extract of *P. cubeba* L. was performed by mixing 18.00 g of the extract with 100 mL methanol and 200 mL water (ratio of 1:2). The methanol–water mixture was added to a one-liter separating funnel, followed by adding 500 mL of n-hexane. The solution was gently swirled to obtain two layers. Hexane, which was the top layer, was removed while the aqueous methanol layer was left in the funnel. An additional 500 mL of n-hexane was added, and the process was repeated until the fraction became light in color. This process was repeated using dichloromethane, ethyl acetate, and n-butanol. All derived fractions, including the aqueous methanol fraction were evaporated to dryness, freeze-dried, and subjected to an antibacterial assay.

### 3.5. Isolation of Compounds by Chromatographic Techniques

#### 3.5.1. Thin Layer Chromatography (TLC)

Thin layer chromatography is routinely used to detect and monitor the various compounds present in the extracts and fractions collected in column chromatography. TLC was carried out on TLC aluminum sheets that had been recoated with silica gel 60 F_254_ (0.25 mm thick. Merck). Extracts or fractions were spotted onto the TLC plates using a capillary tube and developed in a TLC tank saturated with a suitable solvent system at room temperature to separate the components into a wide range of R_f_ values. The components were viewed under ultraviolet light at 254 and 366 nm, respectively. The reagent used for detection was 10% sulfuric acid and the acid was used to stain the spots and detect the presence of a certain type of natural compound by spraying the solution on the TLC plate and heating it at 100 °C for 3–6 min. Stained spots indicate the presence of a particular natural product(s) [31].

#### 3.5.2. Column Chromatography

The chemical compounds in the DCM fractions of the plant were separated over column chromatography based on polarity and molecular weight. Column chromatography was performed using normal phase (NP) silica gel (Merck silica gel 60 PF_254_ No. 7734 (70–230 mesh ASTM) and Merck 9385 (230–400 mesh ASTM). The mobile phase of the column depends on the type of adsorbent/stationary phase used. In all cases, the sample was loaded as a concentrated solution or was pre-adsorbed onto silica gel 230–400 mesh on the top of the adsorbent column.

#### 3.5.3. Spectroscopic Data of Compounds **1** and **2**

##### β-asarone

Compound (**1**) (10.4 mg; yellow tablet crystal): Melting Point: 99–89 °C, MS [M + H]^+^: C_12_H_16_O_3_
*m*/*z* 208. ^1^H-NMR (500 MHz, CDCl3): δ 6.49 (1H, s, H-3), 6.94 (1H, s, H-6), 6.66 (1H, s, H-7), 6.13 (1H, m, H-8), 1.89 (3H, dd, J = 8.0, 8.5Hz, H-9), 3.86 (3H, s, H-10), 3.88 (3H, s, H-11), 3.82 (3H, s, H-12); ^13^C-NMR (125 MHz, CDCl3): δ 97.8 (C-3), 109.6 (C-6), 125.0 (C-7), 124.4 (C-8), 18.8 (C-9), 56.4 (C-10), 56.1 (C-11), 56.7 (C-12), 143.3 (C-1), 148.7 (C-2), 150.6 (C-4), and 118.9 (C-5).

##### Asaronaldehyde

Compound (**2**) (10.8 mg; colorless tablet crystal): Melting Point: 110–112 °C [lit. m.p 110–112 °C, [4], (MS [M + H]^+^: C_10_H_12_O_4_
*m*/*z* 196. ^1^H-NMR (500 MHz, CDCl_3_): δ 6.49 (1H, s, H-3), 7.33 (1H, s, H-6), 10.32 (1H, s, H-7), 3.97 (3H, s, H-8), 3.92 (3H, s, H-9), 3.88 (3H, s, H-10); ^13^C-NMR (125 MHz, CDCl_3_): δ 96.1 (C-3), 109.2 (C-6), 188.3 (C-7), 56.46 (C-8), 56.49 (C-9), 56.56 (C-10), 117.5 (C-1), 143.5 (C-2), 158.9 (C-4), and 156.8 (C-5).

### 3.6. Antibacterial Activity Assay

#### 3.6.1. Sample Preparation

The stock extract of methanol was primed by dispersing a crude extract of *P. cubeba* L. in 100% dimethyl sulfoxide (DMSO; Fisher Scientific, Leicestershire, United Kingdom) to obtain a 100 mg/mL concentration. Further dilution of the solution was done using 1:10 (*v*/*v*) sterile deionized distilled water (ddH_2_O) to produce a 10 mg/mL stock solution. The stock extract was put in storage at 4 °C up to the time it was ready for use. The final concentration of 10% DMSO used in the present study was not effective in killing the tested microorganisms.

#### 3.6.2. Disc Diffusion Assay (DDA)

The compounds isolated from *P. cubeba* L. dichloromethane fraction were screened for antimicrobial activity using the disc diffusion method described by the Clinical and Laboratory Standards Institute [32]. *B. cereus*, *B. subtilis*, *B. pumilus*, and *B. megaterium* were streaked on MHA plates using a sterile cotton swab. A 6-mm sterile filter paper disc was placed on top of the agar and 10 µL of 1 mg/mL (*w*/*v*) compounds were loaded on the paper discs. Chlorhexidine (CHX) in a 0.1% concentration was used as a positive control in the assay. The plates were incubated at 30 °C for 24 h. The presence of a clear zone indicates the inhibition of bacterial growth and the diameter of the zone was measured in millimeters.

#### 3.6.3. Determination of Minimum Inhibitory Concentration (MIC) and Minimum Bactericidal Concentration (MBC)

MIC and MBC were determined using the methods described by CLSI [32]. The MICs and MBCs of the compounds isolated from the *P. cubeba* L. dichloromethane fraction against the vegetative cells of *B. cereus*, *B. subtilis*, *B. pumilus*, and *B. megaterium* were determined using a 96 Wells microtiter plate and two-fold standard broth microdilution methods with an inoculum of approximately 10^6^ CFU/mL. One hundred microliters of compounds stock solution (1 mg/mL = 1000 µg/mL) was mixed and diluted in two-folds with the test organism in MHB (100 µL). Column 12 of the microtiter plate contained the highest concentration of the extract (500 µg/mL) while column 3 contained the lowest concentration (1.95 µg/mL). Column 2 served as the positive growth control for all samples (only MHB and inoculum) while column 1 was the negative growth control (only MHB, no inoculum, and antibacterial agent). The microtiter plate was incubated aerobically at 30 °C for 24 h. MIC is the lowest concentration of antibacterial agent that completely inhibits visible growth. MBC for each bacterial species was determined as outlined for MIC by removing the media from each well that showed no visible growth and subculturing them on MHA plates. The plates were then incubated at 30 °C for 24 h until visible growth was observed in the control plates. Similarly, MBC is defined as the equivalent concentrations required to completely kill microorganisms. Both the MIC and MBC test were done in triplicate.

### 3.7. Antispores Assay

The sporicidal activity of the compounds isolated from *P. cubeba* L. dichloromethane fraction was determined as described by Kida et al. [33] and Rukayadi et al. [27] with some modification. The prepared spores suspension was thawed and diluted at 1:100 in 0.85% NaCl solution (pH 6.6) to obtain initial *B. cereus*, *B. subtilis*, *B. pumilus*, and *B. megaterium* spore suspensions of 2 × 10^6^, 3 × 10^6^, 2.5 × 10^6^, and 1.6 × 10^6^ spores/mL, respectively. The 1% stock extract was diluted in an adjusted spore suspension to obtain the final concentrations of extract (0.03%, 0.06%, 0.125%, 0.25%, and 0.5%). A standard commercially available 25% glutaraldehyde solution (Merck Darmstadt, Germany) was used as a positive control in the determination of sporicidal activity. The glutaraldehyde was diluted at 1:25 in distilled water to yield 1% concentration. The addition of the extract or glutaraldehyde did not change the pH of the test solutions. One milliliter of each concentration was incubated in a 30 °C water bath for varying periods of 0, 1, 2, 3, and 4 h.

An aliquot in the amount 100 μL was transferred to microcentrifuge tubes and centrifuged at (12,000 × *g* at 4 °C for 5 min), and was then rinsed twice with 0.9 mL of 0.85% NaCl solution (pH 6.6) to obtain bacteria-free spores and avoid the effect of vegetative cell residues. The pellets were suspended in 100 μL of 0.85% NaCl solution (pH 6.6), serially diluted, spread onto NA plates, and incubated at 30 °C for 24 h or longer (until the colonies are visible on the plates). The colonies on the duplicate plates were counted and the mean of the colony-forming unit (CFU/mL) was calculated. The differences were obtained by subtracting the log CFU/mL value of the test solution from those of the control (no antibiotic added). The reduction of spore cells in the CFU is expressed as sporicidal activity; the determination of sporicidal activity was repeated twice in triplicate (*n* = 2 × 3).

### 3.8. Statistical Analysis

Excel (v. 2010), and Graph Pad Prism version 6.00 for Windows (v. 6.00, Graph Pad Software, San Diego, CA, USA) were employed to perform the statistical analysis. Results were given as a mean of three replicates ± SD. The significant difference at *p* < 0.05 was established by performing ANOVA.

## 4. Conclusions

This is the first report to describe the isolation of compounds from dichloromethane and *n*-hexane fractions of methanolic extract of *P. cubeba* L berries. In the present study, only two compounds were isolated from the DCM soluble fraction, however, in future research, other active compounds could be isolated such as lignans compounds used to add value to this non-chemically synthesized *P. cubeba* L. extract. The structure of these compounds was identified through several sequences of chromatographic techniques for separation and structure elucidations, i.e., ^1^H-NMR, ^13^C-NMR, 2D NMR, and MS. These compounds were β-asarone 1 and asaronaldehyde 2. The isolated compounds were tested for their antibacterial and antispore activities. In summary, it is remarkable to note that β-asarone 1 and asaronaldehyde 2 conferred significant antibacterial and antispore activities against the vegetative cells and spores of *B. cereus* ATTC33019, *B. subtilis* ATCC6633, *B. pumilus* ATCC14884, and *B. megaterium* ATCC14581 [34]. Thus, β-asarone 1 and asaronaldehyde 2 might be good to develop as a food preservative. However, the mechanisms of killing require further investigation

Moreover, the combination of β-asarone with asaronaldehyde and the other compounds that were identified could be tested to determine any synergistic effects that might increase the antibacterial and sporicidal activity. Several studies have been reported on the potential of this synergistic effect with the aim of improving the biological activity of some low-activity plant extracts or compounds by combining them with higher activity plants or combining lower activity compounds with higher activity compounds.

## Figures and Tables

**Figure 1 molecules-24-03095-f001:**
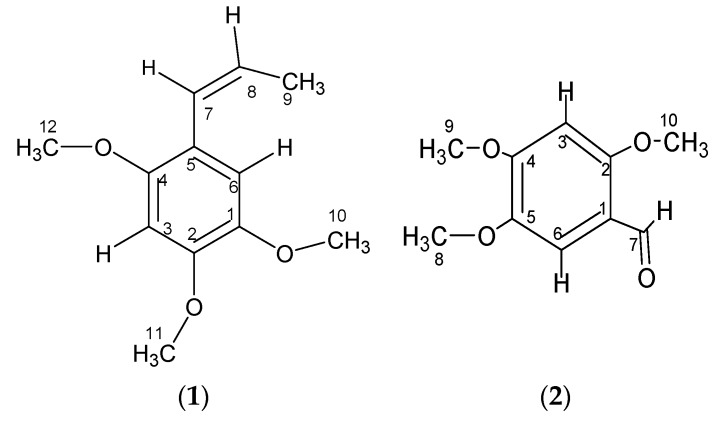
Structures of the compounds isolated from *Piper cubeba* L. berries.

**Figure 2 molecules-24-03095-f002:**
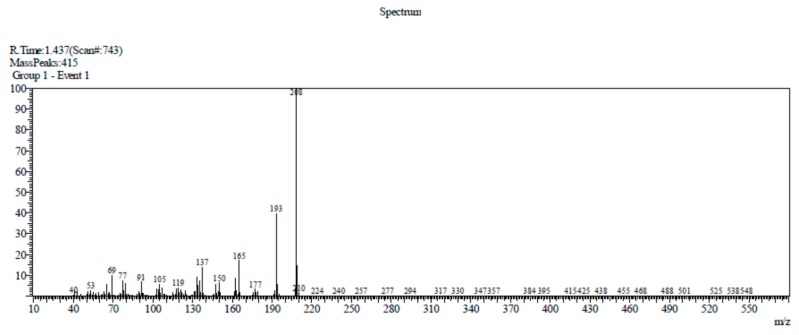
Direct injection probe mass spectrometer (DIP-MS) spectrum of β-asarone.

**Figure 3 molecules-24-03095-f003:**
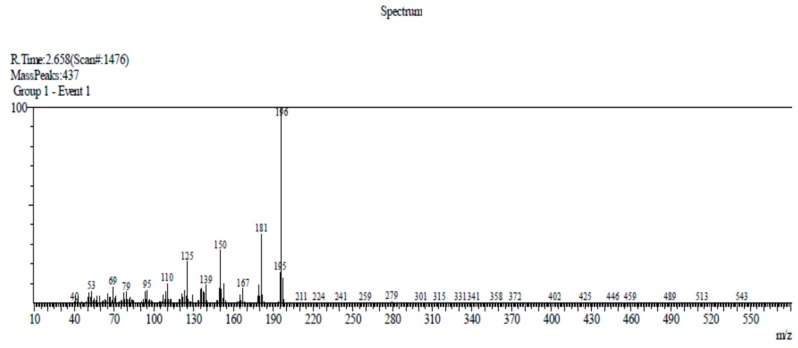
DIP-MS spectrum of asaronaldehyde.

**Table 1 molecules-24-03095-t001:** Disc diffusion of isolated compounds against *Bacillus* sp.

*Bacillus* sp.	Inhibition Zone (mm) ± SD			
	AA	β-A	CHX	DMSO
*B. cereus* ATCC33019	8.00 ± 0.00	9.31 ± 0.50	11.4 ± 0.31	n.a
*B. subtilis* ATCC6633	8.81 ± 0.24	8.51 ± 0.50	11.1 ± 0.10	n.a
*B. pumilus* ATCC14884	7.81 ± 0.22	8.51 ± 0.50	11.1 ± 0.10	n.a
*B. megaterium* ATCC14581	7.21 ± 0.21	8.15 ± 0.50	11.6 ± 0.30	n.a

n.a: No activity. The diameter of the inhibition zones is in mm (including disc). Positive control (Chlorhexidine: CHX; 0.1%). Negative control (DMSO; 10%). Results were expressed as means ± standard deviation (SD); *n* = 2 × 3. AA: Asaronaldehyde, and β-A: β-asarone.

**Table 2 molecules-24-03095-t002:** Minimum inhibitory concentrations (MICs) and minimum bactericidal concentrations (MBCs) of β-asarone, and asaronaldehyde against *B. cereus*, *B. subtilis*, *B. megaterium*, and *B. pumilus*.

*Bacillus* sp.	Concentration of Compounds (µg/mL)			
	Asaronaldehyde		β-Asarone	
	MICs	MBCs	MICs	MBCs
*B. cereus* ATCC33019	125	500	125	250
*B. subtilis* ATCC6633	125	500	125	250
*B. pumilus* ATCC14884	125	500	125	250
*B. megaterium* ATCC14581	125	500	125	250

**Table 3 molecules-24-03095-t003:** The sporicidal activity of β-asarone against the spores of *B. cereus* ATCC33019.

Concentration (µg/mL (% *w*/*v*))	Time (h)			
	1	2	3	4
0.00	^a^ 6.17 ± 0.15	^a^ 6.17 ± 0.15	^a^ 6.17 ± 0.15	^a^ 6.17 ± 0.15
62.5	^b^ 5.87 ± 0.02	^b^ 5.58 ± 0.02	^b^ 5.35 ± 0.03	^b^ 5.13 ± 0.05
125	^c^ 4.63 ± 0.03	^c^ 4.51 ± 0.02	^c^ 4.42 ± 0.02	^c^ 4.30 ± 0.02
250	^c^ 4.12 ± 0.02	^c^ 4.10 ± 0.10	^d^ 3.98 ± 0.02	^d^ 3.87 ± 0.02
500	^d^ 3.88 ± 0.01	^d^ 3.67 ± 0.01	^d^ 3.48 ± 0.04	^d^ 3.25 ± 0.07
1000	^d^ 3.08 ± 0.23	^e^ 2.74 ± 0.04	^e^ 0.00 ± 0.00	^e^ 0.00 ± 0.00

Results are presented as mean ± standard deviation. Significant differences in means (*n* = 2 × 3). Within the same column is indicated with different letters (*p* < 0.05).

**Table 4 molecules-24-03095-t004:** The sporicidal activity of β-asarone against the spores of *B. subtilis* ATCC6633.

Concentration (µg/mL (% *w*/*v*))	Time (h)			
	1	2	3	4
0.00	^a^ 6.24 ± 0.05	^a^ 6.24 ± 0.05	^a^ 6.24 ± 0.05	^a^ 6.24 ± 0.05
62.5	^b^ 5.90 ± 0.04	^b^ 5.61 ± 0.06	^b^ 5.37 ± 0.03	^b^ 5.18 ± 0.07
125	^c^ 4.66 ± 0.02	^c^ 4.55 ± 0.05	^c^ 4.47 ± 0.03	^c^ 4.33 ± 0.04
250	^c^ 4.17 ± 0.02	^c^ 4.16 ± 0.05	^d^ 3.99 ± 0.10	^d^ 3.90 ± 0.02
500	^d^ 3.88 ± 0.01	^d^ 3.71 ± 0.05	^d^ 3.52 ± 0.06	^d^ 3.25 ± 0.13
1000	^d^ 3.15 ± 0.40	^e^ 2.81 ± 0.05	^e^ 0.00 ± 0.00	^e^ 0.00 ± 0.00

Results are presented as mean ± standard deviation. Significant differences in means (*n* = 2 × 3). Within the same column is indicated with different letters (*p* < 0.05).

**Table 5 molecules-24-03095-t005:** The sporicidal activity of β-asarone against spores of *B. pumilus* ATCC14884.

Concentration (µg/mL (% *w*/*v*))	Time (h)			
	1	2	3	4
0.00	^a^ 6.37 ± 0.04	^a^ 6.37 ± 0.04	^a^ 6.37 ± 0.04	^a^ 6.37 ± 0.04
62.5	^b^ 5.80 ± 0.04	^b^ 5.50 ± 0.06	^b^ 5.44 ± 0.04	^b^ 5.26 ± 0.05
125	^c^ 4.68 ± 0.02	^c^ 4.55 ± 0.05	^c^ 4.47 ± 0.02	^c^ 4.33 ± 0.04
250	^c^ 4.17 ± 0.02	^c^ 4.16 ± 0.05	^d^ 3.99 ± 0.10	^d^ 3.90 ± 0.02
500	^d^ 3.88 ± 0.01	^d^ 3.71 ± 0.05	^d^ 3.53 ± 0.07	^d^ 3.21 ± 0.07
1000	^d^ 3.51 ± 0.15	^d^ 3.43 ± 0.05	^e^ 0.00 ± 0.00	^e^ 0.00 ± 0.00

Results are presented as mean ± standard deviation. Significant differences in means (*n* = 2 × 3). Within the same column is indicated with different letters (*p* < 0.05).

**Table 6 molecules-24-03095-t006:** The sporicidal activity of β-asarone against spores of *B. megaterium* ATTC14581.

Concentration (µg/mL (% *w*/*v*))	Time (h)			
	1	2	3	4
0.00	^a^ 6.32 ± 0.04	^a^ 6.32 ± 0.04	^a^ 6.32 ± 0.04	^a^ 6.32 ± 0.04
62.5	^b^ 5.70 ± 0.04	^b^ 5.65 ± 0.06	^b^ 5.41 ± 0.04	^b^ 5.12 ± 0.05
125	^c^ 4.67 ± 0.02	^c^ 4.50 ± 0.05	^c^ 4.43 ± 0.02	^c^ 4.23 ± 0.04
250	^c^ 4.13 ± 0.02	^c^ 4.11 ± 0.05	^d^ 3.89 ± 0.10	^d^ 3.40 ± 0.02
500	^d^ 3.81 ± 0.01	^d^ 3.67 ± 0.04	^d^ 3.33 ± 0.07	^d^ 3.11 ± 0.07
1000	^d^ 3.41 ± 0.15	^d^ 3.14 ± 0.04	^e^ 0.00 ± 0.00	^e^ 0.00 ± 0.00

Results are presented as mean ± standard deviation. Significant differences in means (*n* = 2 × 3). Within the same column is indicated with different letters (*p* < 0.05).

**Table 7 molecules-24-03095-t007:** The sporicidal activity of asaronaldehyde against the spores of *B. cereus* ATCC33019.

Concentration (µg/mL (% *w*/*v*))	Time (h)			
	1	2	3	4
0.00	^a^ 5.70 ± 0.13	^a^ 5.70 ± 0.13	^a^ 5.70 ± 0.13	^a^ 5.70 ± 0.13
62.5	^b^ 4.58 ± 0.08	^b^ 4.53 ± 0.07	^b^ 4.45 ± 0.09	^b^ 4.32 ± 0.05
125	^b^ 4.52 ± 0.09	^b^ 4.41 ± 0.15	^b^ 4.29 ± 0.09	^b^ 4.18 ± 0.12
250	^b^ 4.47 ± 0.15	^b^ 4.35 ± 0.19	^b^ 4.23 ± 0.17	^b^ 4.05 ± 0.15
500	^c^ 3.40 ± 0.21	^c^ 3.20 ± 0.09	^c^ 3.00 ± 0.00	^c^ 0.00 ± 0.00
1000	^d^ 0.00 ± 0.00	^d^ 0.00 ± 0.00	^d^ 0.00 ± 0.00	^c^ 0.00 ± 0.00

Results are presented as mean ± standard deviation. Significant differences in mean (*n* = 2 × 3). Within the same column is indicated with different letters (*p* < 0.05).

**Table 8 molecules-24-03095-t008:** The sporicidal activity of asaronaldehyde against spores of *B. subtilis* ATCC6633.

Concentration (µg/mL (% *w*/*v*))	Time (h)			
	1	2	3	4
0.00	^a^ 6.39 ± 0.12	^a^ 6.39 ± 0.12	^a^ 6.39 ± 0.12	^a^ 6.39 ± 0.12
62.5	^b^ 4.66 ± 0.02	^b^ 4.58 ± 0.02	^b^ 4.49 ± 0.06	^b^ 4.26 ± 0.10
125	^b^ 4.61 ± 0.09	^b^ 4.54 ± 0.07	^b^ 4.50 ± 0.12	^b^ 4.37 ± 0.05
250	^b^ 4.58 ± 0.07	^b^ 4.33 ± 0.04	^b^ 4.14 ± 0.04	^c^ 3.93 ± 0.05
500	^b^ 4.30 ± 0.08	^c^ 3.80 ± 0.11	^c^ 3.60 ± 0.09	^c^ 3.58 ± 0.07
1000	^c^ 3.83 ± 0.32	^c^ 3.40 ± 0.09	^c^ 3.10 ± 0.08	^d^ 0.00 ± 0.00

Results are presented as mean ± standard deviation. Significant differences in mean (*n* = 2 × 3). Within the same column is indicated with different letters (*p* < 0.05).

**Table 9 molecules-24-03095-t009:** The sporicidal activity of asaronaldehyde against spores of *B. pumilus* ATCC14884.

Concentration (µg/mL (% *w*/*v*))	Time (h)			
	1	2	3	4
0.00	^a^ 5.48 ± 0.04	^a^ 5.48 ± 0.04	^a^ 5.48 ± 0.04	^a^ 5.48 ± 0.04
62.5	^b^ 4.77 ± 0.11	^b^ 4.69 ± 0.06	^b^ 4.59 ± 0.07	^b^ 4.47 ± 0.08
125	^b^ 4.72 ± 0.06	^b^ 4.63 ± 0.05	^b^ 4.55 ± 0.05	^b^ 4.38 ± 0.04
250	^b^ 4.63 ± 0.05	^b^ 4.52 ± 0.04	^b^ 4.40 ± 0.04	^b^ 4.28 ± 0.10
500	^b^ 4.00 ± 0.80	^c^ 3.90 ± 0.09	^c^ 3.75 ± 0.50	^c^ 3.60 ± 0.60
1000	^c^ 3.30 ± 0.80	^c^ 3.00 ± 0.00	^d^ 0.00 ± 0.0	^d^ 0.00 ± 0.00

Results are presented as mean ± standard deviation. Significant differences in mean (*n* = 2 × 3). Within the same column is indicated with different letters (*p* < 0.05).

**Table 10 molecules-24-03095-t010:** The sporicidal activity of asaronaldehyde against spores of *B. megaterium* ATCC14581.

Concentration (µg/mL (% *w*/*v*))	Time (h)			
	1	2	3	4
0.00	^a^ 5.31 ± 0.04	^a^ 5.31 ± 0.04	^a^ 5.31 ± 0.04	^a^ 5.31 ± 0.04
62.5	^b^ 4.50 ± 0.17	^b^ 4.41 ± 0.18	^b^ 4.34 ± 0.18	^b^ 4.24 ± 0.21
125	^b^ 4.43 ± 0.10	^b^ 4.33 ± 0.15	^b^ 4.23 ± 0.22	^b^ 4.15 ± 0.20
250	^b^ 4.39 ± 0.09	^b^ 4.24 ± 0.18	^b^ 4.10 ± 0.18	^b^ 3.94 ± 0.28
500	^c^ 3.90 ± 0.08	^c^ 3.80 ± 0.09	^c^ 3.58 ± 0.12	^c^ 3.36 ± 0.14
1000	^c^ 3.20 ± 0.90	^c^ 3.00 ± 0.00	^d^ 0.00 ± 0.00	^d^ 0.00 ± 0.00

Results are presented as mean ± standard deviation. Significant differences in mean (*n* = 2 × 3). Within the same column is indicated with different letters (*p* < 0.05).

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
