# Peer review of "Antibacterial and Antispore Activities of Isolated Compounds from Piper cubeba L."

_molecules, 2019, doi:10.3390/molecules24173095_

Round 1
Reviewer 1 Report
The yields of the two compounds i.e. beta-asarone and asaronaldehyde should be given.
Language polishing is necessary. There are places throughout the manuscript that require correction e.g. line 19 should read "were" instead of "will be". Line 130 should read"corresponded".Line 152should read "results showed".
Data disclosing the mechanisms of antibacterial and antispore/sporicidal activity of the two compounds i.e. beta-asarone and asaronaldehyde should be provided.
Author Response
The manuscript have been revised as suggested

Reviewer 2 Report
This manuscript describe the antibacterial and antispore activities of natural compounds isolated from piper cubeba L. There are good analytical work and use of instrument to characterize the active compounds. This is well written manuscript and has good research idea, I have few comments to consider: At the end of introduction, the objective statement need to be revised, the word chapter should be study or work. It seems that there are 2 objectives not one objective. Under materials and methods, the first paragraph can be omitted. Under the conclusion, can you include some limitations for this study and future direction, what are the possible food applications and what possible studies in foods? Incorporation of the natural compounds to study the shelf-life extension and safety of local food products?
Author Response

(The authors gave the same response as above.)

Round 2
Reviewer 1 Report
The manuscript is Acceptable in present form